# Stability and Dynamic Walk Control of Humanoid Robot for Robot Soccer Player

**Rudolf Jánoš** [1,*], **Marek Sukop** [1], **Ján Semjon** [1], **Peter Tuleja** [1], **Peter Marcinko** [1], **Martin Kočan** [1], **Maksym Grytsiv** [1], **Marek Vagaš** [2], **Ľubica Miková** [2] **and Tatiana Kelemenová** [3]

1 Department of Production Technology and Robotics, Faculty of Mechanical Engineering, Technical University of Kosice, 04200 Kosice, Slovakia; marek.sukop@tuke.sk (M.S.); jan.semjon@tuke.sk (J.S.); peter.tuleja@tuke.sk (P.T.); peter.marcinko@tuke.sk (P.M.); martin.kocan@student.tuke.sk (M.K.); maksym.grytsiv@tuke.sk (M.G.)
2 Department of Industrial Automation and Mechatronics, Faculty of Mechanical Engineering, Technical University of Kosice, 04200 Kosice, Slovakia; marek.vagas@tuke.sk (M.V.); lubica.mikova@tuke.sk (Ľ.M.)
3 Department of Biomedical Engineering and Measurement, Faculty of Mechanical Engineering, Technical University of Kosice, 04200 Kosice, Slovakia; tatiana.kelemenova@tuke.sk
* Correspondence: rudolf.janos@tuke.sk; Tel.: +421-55-6022197

**Abstract:** Robotic football with humanoid robots is a multidisciplinary field connecting several scientific fields. A challenging task in the design of a humanoid robot for the AndroSot and HuroCup competitions is the realization of movement on the field. This study aims to determine a walking pattern for a humanoid robot with an impact on its dynamic stability and behavior. The design of the proposed technical concept depends on its stability management mechanism, walking speed and such factors as the chosen stability approaches. The humanoid robot and its versatility, along with the adaptability of the terrain, are somewhat limited due to the complexity of the walking principle and the control of the robot's movement itself. The technical concept uses dynamic stability as the potential force of the inertial bodies and their parts so that the humanoid robot does not overturn. The total height of the robot according to the rules of the competition will be 50 cm. In the performed experiment, only the lower part of the humanoid robot with added weight was considered, which is more demanding due to the non-use of the upper limbs for stabilization. The performed experiment verified the correctness of the design, where the torso of the robot performed eight steps in inclinations of a roll angle $+4/-2°$ and a pitch angle $+4/-6°$.

**Keywords:** humanoid robot; walking pattern; stability

## 1. Introduction

In general, a humanoid robot is statically stable if its stability is maintained at any point in time of its movement [1]. Static stability is secured in the condition where the projection of the robot's center of gravity is at all times within the convex polygon defined by the feet that are currently touching the pad. If the construction of the humanoid robot is less than three feet long, the support polygon degenerates to a line or a point. In such a situation, the system is dynamically stable, provided it is in balance. If static stability has to be always maintained, the system will be severely limited in speed and maneuverability.

Humanoid robots are statically unstable; however, they become dynamically stable at a moderate speed [2]. Dynamic stability increases with increasing speed, and it is always necessary for bipedal (there are examples of bipedal robots that do not require dynamic stability if the robot walks slowly) and one-legged robots, but not required for multi-legged robots.

One of the main problems is the continuous and dynamically balanced creation of a walking pattern. There are many methods and techniques to overcome this requirement, such as:

The inverted pendulum linear model (LIPM) is one of the solution techniques. The bipedal robot is treated as a simple inverted pendulum problem. If the robot trunk starts to fall while walking, its support leg prevents it from falling.

Another popular approach to the dynamic stability method to create a walking pattern is the zero-moment criterion (ZMP). ZMP is defined as the point on earth where the sum of the moments of active forces equals zero. If the ZMP is inside the convex torso of all contact points between the feet and the ground, a dynamically stable walking movement can be achieved [3,4].

A similar problem with gait generation was solved by S. Kajita, who designed the ZMP tracking servo controller. This controller was used to compensate for the ZMP error caused by the difference between the model and the multi-body system [5]. Another approach to dynamic gait control was chosen by J.H. Kim, who controls the humanoid robot, as a simple model of a reversible pendulum with a complaint joint. The humanoid robot is equipped with a comprehensive sensor system that provides quality feedback to maintain kinematic stability [6].

Dynamic stability is very advantageous and desirable in terms of speed. However, dynamic stability is much more complicated to control since it is necessary to permanently monitor the movement of all legs for fulfilment of this condition [7]. For ensuring this stability, it is also necessary to perfectly know the mechanical and structural characteristics of humanoid robotic system and its chassis structure, as well as the distribution of weight throughout its subsystem. The biological system of a human can be considered as a model for a two-legged walking robotic system [8].

As a contribution of this work, a method of generating a walking pattern for a robot soccer player was implemented in this study. Simple sensor systems are used for the design of a robotic footballer due to the footballer's resistance in contact play as well as for economic reasons.

The main benefit of this paper is a formal method for determining the parameters of a human-inspired controller, which results in a demonstrably stable robotic gait that is "as human-like as possible" and robust enough for the robot to maintain stability even when the robot is affected by other environmental influences. Much of our work focuses on walking principles in humanoid robotics because it presents a challenging environment with measurable functional outcomes [9].

Forces applied to robot movements are critical in creating a smooth walking process that is firm enough to hold but does not break its stability or falling. Early work with computer-enhanced robotic stability represents a field in its infancy. The evidence strongly suggests that the ability to confer quality feedback to present efficient robotic systems would contribute significantly to the safe performance of the whole procedures with these complex systems. The lack of quality feedback in current robotic systems is a significant handicap in performing technically more complex moving maneuvers and maintaining stability for the implementation of walking in indoor and outdoor applications.

From our own observations of experienced and talented humanoid robots during their training with various walking principles broken, delicate algorithms are often torn due to the application of excessive forces conventionally attenuated with insufficient feedback control. The consequences of walking errors or excessive delay with the walking process present much greater potential for irreversible falling or even damages of the robot construction.

A common refrain in the current robotic literature with respect to the walking applicability of these robotic systems includes a rather steep learning curve and the lack of control as major restrictions. At the beginning, there is a learning principle, and one has to proceed very wisely, but the technology can be learned. The only limitation for the moment is the loss of sufficient feedback [10].

After a short introduction, in Section 2 we focused on the possibilities of solving the static and dynamic stability of a two-legged robot. In Section 3 we focused on the description of the robot kinematics and the description of the performed experiment.

The experimental results are presented in Section 4. Sections 5 and 6 are devoted to the discussion and conclusion.

## 2. Stability

Current approaches for solving of stability problem at biped-on walking robots are focused on assumption of two different supports i.e., support planes [11]. These support planes are oriented in any direction in space and subordinated to moments $M_L$ and $M_R$. The forces of these moments act from the pad to the left $L_L$ and right $L_R$ leg, and they are perpendicular to their support surfaces in Figure 1. Similarly, hands are objects of the acting $Q_L$ and $Q_R$ moments arising from forces $q_L$ and $q_R$ acting on left and right hand. These moments are not perpendicular to any surface [12].

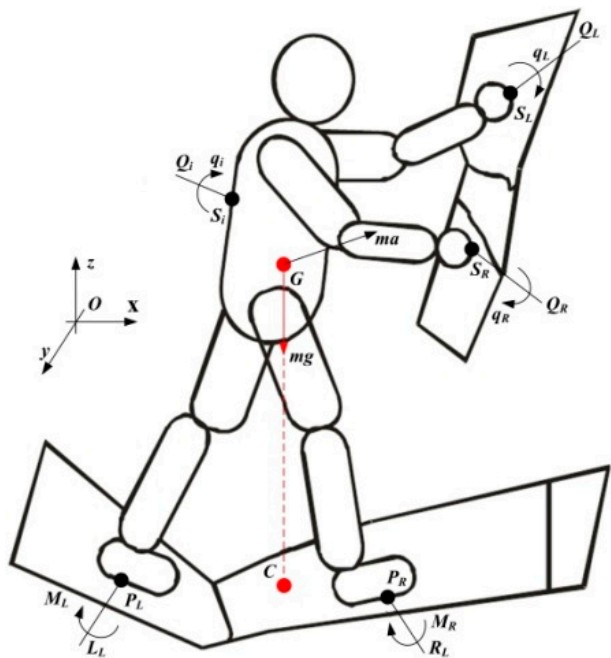

**Figure 1.** Acting forces and moments to humanoid robot.

Regarding this, we assume that a humanoid robot will be involved in real-world tasks and will thus inevitably be the object of several anticipated, but also unexpected, forces and moments from the environment. Therefore, we can state the acting effect set of m-forces $qi$ to arbitrary points of $Si$ and set of p-moments $Qi$.

The robot is created by $n$—segments $L_i$ with weight $m_i$, which are located in the center of mass $G_i$. The entire robot weight—$m$ is as follows:

$$m = \sum_L^n m_i,$$ (1)

And is situated to the centre of mass (CoG). Then, the formula for dynamic humanoid robot balance can be written as:

$$L_L + L_R + \sum_{i=1}^n m_i g + \sum_{i=1}^m Q_i = \sum_{i=1}^n m_i a_i,$$ (2)

where $a_i$—is acceleration of $i$th segment; $L_L$—force that is created during left leg contact with the ground and acts at point $P_L$; $L_R$—force that is created during right leg contact with the ground at point $P_R$; $Q_L$—force that is created during left hand contact and acts at point $S_L$; and $Q_R$—force that is created during right hand contact and acts at point $S_R$.

Equation (2) is written as follows:

$$L^c + Q = ma,$$ (3)

where $a$—acceleration CoG; $L_L + L_R = L_C$—resulting force that acts from ground to legs; and $Q = \sum Q_i$—resulting external force that acts from hands.

In the case of disturbing external forces neglecting $Q$ that act to the humanoid robot, we obtain a balance of external forces and gravitational–inertia forces:

$$Q + L^C = -L^{gi} \tag{4}$$

$$L^{gi} = m(g - a). \tag{5}$$

The equation for the moment calculation at arbitrary inertial reference point $O$ is:

$$
\begin{aligned}
M_L + M_R + OP_L \times L_L + OP_R \times L_R + \sum_{i=1}^{p} q_i + \sum_{i=1}^{m} OS_i \times Q_i \\
+ \sum_{i=1}^{n} OG_i \times m_{ig} = \sum_{i=1}^{n} \dot{H}_O = \sum_{i=1}^{n} \dot{H}_{G_i} + \sum_{i=1}^{n} OG_i \times m_i a_i,
\end{aligned} \tag{6}
$$

where $H_{Gi}$ is the centric angular momentum of $i$th segment at point $G_i$ and defined as:

$$H\dot{\,}\_(G\_i) = L\_i\,(I\_i\,\omega\dot{\,}\_i - (I\_i\,\omega\_i) \times \omega\_i), \tag{7}$$

where $L_i$—rotation matrix of ith segment; $I_i$—inertia matrix; $\omega_i$—speed of rotation; $\dot{\omega}_i$—angular acceleration.

The centric angular momentum of the humanoid robot is defined at CoG and determined by the formula as follows:

$$\dot{H}_{G_i} = \sum \dot{H}_{G_i} + \sum GG_i \times m_i a_i. \tag{8}$$

Equation (6) can be adopted to the shape:

$$M + OP_L \times L_L + OP_R \times L_R + q + \sum_{i=1}^{m} OS_i \times Q_i + OG \times mg = \dot{H}_O = \dot{H}_{G_i} + OG \times ma, \tag{9}$$

where $M = M_L + M_R$, $q = \sum q_i$.

The determination of lateral moments created from external noise can be considered as the equilibrium of two opposite moments: the moment caused by contact with the ground on one side and gravity and inertia on the other side:

$$
\begin{aligned}
M_O^{ext} + M_O^C &= -M_O^{gi}, \\
M_O^{ext} &:= q + \sum_{i=1}^{m} OS \times Q, \\
M_O^C &:= M + OP_L \times L_L + OP_R \times L_R, \\
M_O^{gi} &:= mOG \times (g - a) - \dot{H}_g.
\end{aligned} \tag{10}
$$

In the case of neglecting of moments around CoG—, Equation (6) can be adopted to:

$$M + OP_L \times L_L + OP_R \times L_R + q + \sum_{i=1}^{m} GS_i \times Q_i = \dot{H}_{G_i}. \tag{11}$$

In a simpler case of a humanoid walking cycle on a flat surface, we can assume:

- The surface is flat, but not absolutely flat;
- Forces and moments (LL/ML, LR/MR) that are created during contact with the ground can be substituted by one overall resulting force (LC = LL + LR) acting during contact with the ground;
- The principle consists of the resulting force, which acts at point P, that is located in the support area, $q = 0$ (resulting moment $M$ created during contact with the ground);
- The value of resulting force $M$ still has the same vector part only in case of overturn and instability of the humanoid robot [13].

Then, we can write the following Formula (9) in a shape:

$$[M+]OP \times L^C + OG \times mg = \dot{H}_O = \dot{H}_G + OG \times ma. \tag{12}$$

Equations (5) and (10) together describe a balance of the humanoid robot and can be written as follows:

$$\begin{aligned} L^C &= -L^{gi}, \\ M_O^C &= -M_O^{gi}, \end{aligned} \tag{13}$$

where

$$\begin{aligned} M_O^C &= -M_O^{gi} \text{ and } L^{gi} = m(g - a), \\ M_O^{gi} &= OG \times L_{gi} - \dot{H}_G. \end{aligned} \tag{14}$$

Neglecting the moments around CoG Equation (12) gives the following shape:

$$[M+]GP \times L^C = \dot{H}_G. \tag{15}$$

In the case of the humanoid robot having a payload about weight $m_L$, if this weight is located at the point and external disturbing force (failure force) $Q$ is located at point $S_Q$, then Equation (9) is written as:

$$[M+]OP \times L^C + OS_L \times m_L g + OS_Q \times Q + OG \times mg = \dot{H}_G + OG \times ma. \tag{16}$$

In the case of neglecting the moment around CoG, the formula can be transformed to a simpler version and in a finalizing shape:

$$[M+]GP \times L^C + GS_L \times m_L g + GS_Q \times Q = \dot{H}_G \tag{17}$$

Obviously, dynamic stability of a humanoid robot is influenced by several force effects [14]. These are generated either from the inside and are given by the weight of humanoid robot structure parts, or from the outside, which are given, for example, by weight or inertia effects. Therefore, many movements of humanoid robot structure parts are used to ensure dynamic stability, so that the centre of gravity can be effectively changed during walking [15].

*Data Collection and Analysis*

Inverse kinematics was applied to obtain the values of individual partial motions of the humanoid robot for walking cycles. Therefore, primary data were collected from the biological movement of the human model by using SMART system (Capture, Tracker). The system is designed for many motion recording applications ranging from clinical to sports or industrial purposes, etc. It is used wherever the trajectory of movement of patients, athletes, actors or other objects has to be measured and recorded. [16] The principle of operation lies in recognition and recording of positions at small passive marks placed on the body at the analysed object. The position of these markers, their speed and acceleration are measured automatically in real time. The system also integrates other devices and sensors that allow advanced data collection to be exported for further analysis in different programs (Matlab etc.) [17]. Since a humanoid robot is not equipped with any sensory subsystems with the aim of collecting dynamic stability information (e.g., methods of CoP, ZMP, ZRAM or FRI), a suitable indicator of stability control for this proposal could be monitoring of the center of mass showing the centre of mass on the ground [18]. The control of the walking cycle is controlled by a combination of dynamic filter and low-level joint measurement. The essence of this principle consists of trajectories tracking of motion paths at individual joints on the humanoid robot by results from motion equations on the basis of created tables with values for rotation angles at individual motors–joints.

### 3. Parameters for the Experiment

#### 3.1. Functional and Dimensional Properties of a Tested Solution

An experimental humanoid robot has 12 degrees of freedom (DOF). The complete kinematic diagram of a skeleton biological model/man has 30° DOF, and this number is reduced only to the inevitably necessary movements. Construction itself would be unsuitable because of its complexity, so it is necessary to re-evaluate input requirements for complexity of movement of the humanoid robot as well as its application requirements [15].

On the basis of these criteria, necessary simplifications of a kinematic scheme were then implemented to meet the application requirements for participation in robotic football and, at the same time, simplicity of design or way of controlling its own motion. Its kinematic system is connected at the site of the pelvic bone with the right and left legs being the means of resilient elements that partially absorb inertia energy of the moving masses between the legs and also damp undesirable oscillation on the movement resulting from the transfer of weight from one foot to the other. The flexibly placed legs with the pelvis are shown in Figure 2.

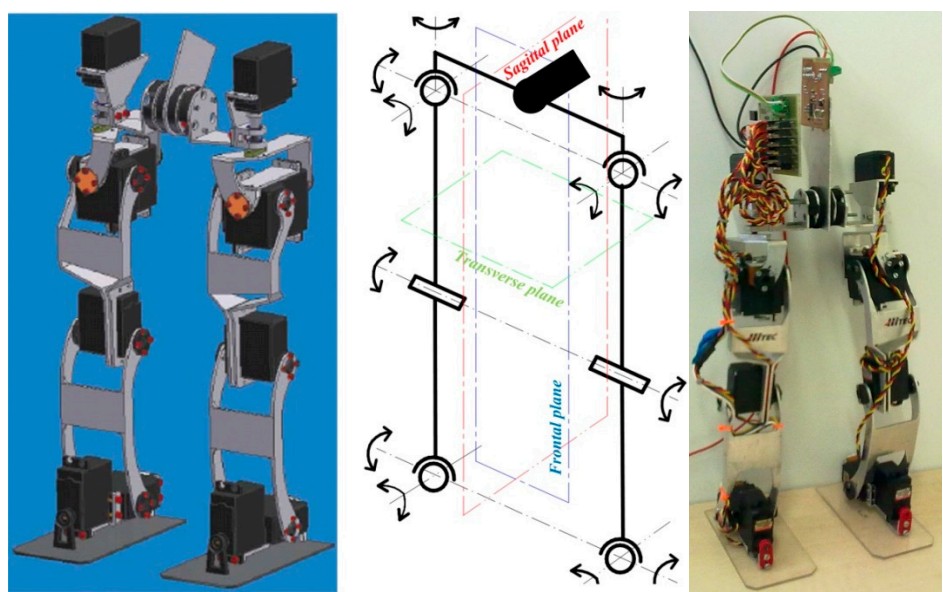

**Figure 2.** Experimental humanoid robot with its kinematics system.

The mobility ranges at individual joints are composed of a complex sequence of complex partial movements and are limited by the robustness of the building elements or size of the drive units. The hip joint has three DOF, the knee joint has one DOF and the ankle has two DOF. In total, the leg of the locomotor system of the bipedal robot has six DOF. The joints of the joints were designed to build on the biological model of a man and allow for brave mobility. The ranges of positions on the humanoid robot are shown in Table 1.

**Table 1.** Ranges of positions on humanoid robot.

| Joint | DOF | Position | Angle |
|---|---|---|---|
| Hip joint | $3 \times 2 = 6$ | extension | up to 120° |
| | | flexion | up to 60° |
| | | abduction | up to 45° (crossing of legs) |
| | | abduction | up to 105° |
| | | vertical rotation (inside to the body) | up to 35° |



**Table 1.** *Cont.*

| Joint | DOF | Position | Angle |
|---|---|---|---|
| Knee joint | 1 × 2 = 2 | flexion | up to 130° with upright body position up to 154° |
| Ankle joint | 2 × 2 = 4 | flexion | with half-squat body position |
| | | supination | up to 32° |
| | | pronation | up to 24° |
| | | extension | up to 120° |
| | | flexion | up to 80° |

Design of the Drive Block Module

The drive units for individual joints of the humanoid robot for walking are based on the servo-drive principle and complemented by incremental encoders. The drive torque is 0.96 Nm with value of 6 V and a weight of 55 g. The control of such a specific drive is realized via an ATMega 168 microprocessor apparatus with PID regulation for speed control regarding the output of shaft [9]. The overall gear ratio $n$ = 286.8, number of increments is 50 and ratio between the drive and pinion of the sensor $p$ = 1.4. The number of revolutions at the sensing wheel during rotation of the output shaft is about 360°:

$$O_{S-360} = \frac{n}{p} = \frac{286.8}{1.4} = 204.83 \; ot/360°. \tag{18}$$

The number of increments per one rotation at the output shaft can be evaluated as follows:

$$INK_{360} = O_{S-360} \times ink = 204.83 \times 50 = 10\,241.5 \frac{ink^°}{360}. \tag{19}$$

The smallest drive step that is possible to reach the aim of proposed sensing and controlling is as follows:

$$Step = \frac{360°}{INK_{360}} = \frac{360}{10\,241.5} = 0.03515°. \tag{20}$$

The selection of drives and calculation of its necessary torques was based on the most unfavorable humanoid robot positions, where drives are mostly heavily loaded. These usually consist of bending at forward positions (overhangs) or other bend positions.

*3.2. Data Processing*

The first step includes the processing and recording of a walking cycle from an analyzed object (human) through the SMART system. After that, obtained 3D data of markers and their positions were implemented into the Matlab software together with input parameters defining the number of sequences, number of using points, start and end of walking cycle and the start and end of the support and walking phase [14,16]. These values are needed for further data processing and for obtaining the trajectories of joint motion and coordinates during the one-step phase.

Trigonometric functions and a cosine sentence were used to calculate the rotation of moving joints. We used the following assumptions to calculate joint rotation:

- Length of step for one leg d = 240 mm;
- During transfer phases of the legs, the ankle together with the foot moves parallel with the ground in an ellipsoid trajectory with a length of 30 mm;
- Height of step v = 30 mm;
- Length of femur and fibula l = 110 mm;
- Distance of hip joints b = 115 mm;
- Starting value for each angle for rotating joints is 90°.

### 3.3. Walking Pattern Generation

Inverse pendulum methodology (IPM) and zero moment point (ZMP) is used during walking pattern generation while we obtain CoG. This method can be used in case the weight of the legs is much smaller than that of the entire walking robot. The entire mass of the trunk, head and (eventually) hands of the walking robot should be concentrated at one point on the robot's trunk—CoG. It is also assumed that the CoG movement is on the horizontal plane. The ZMP Equation in relationship to the CoG position $(c_x, c_y \; c_z = const.)$ can be described from the moment that is created from the gravitational force and from the dynamic moment created by acceleration. For the ZMP position $(p_x, p_y, 0)$, we can describe:

$$p_x = c_x - \frac{c_z}{g} \ddot{c}_x, \tag{21}$$

$$p_y = c_y - \frac{c_z}{g} \ddot{c}_y. \tag{22}$$

As the minimum requirement for the robot's mechanism is to maintain its static stability, the ZMP should be on the foot support area while the moving leg is moving. If both feet are on the ground, the ZMP moves from one foot to the other. This cycle is repeated while the robot is walking, so this can be used to create a walking path [19]. The reference paths of ZMP are described in Figure 3, where the Reference CoG is shown in blue, the Reference ZMP is shown in red and the Fourier approximation of ZMP is shown in black.

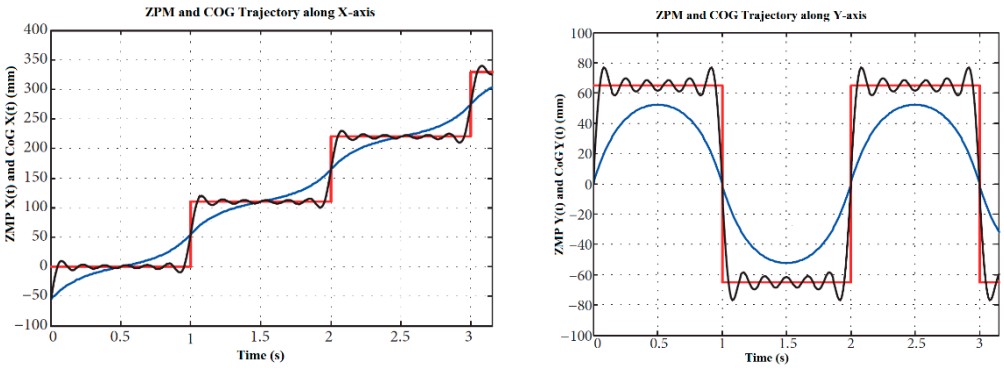

**Figure 3.** Reference trajectory of ZMP, CoG and Fourier transformation of ZMP.

An important note consists of the assumption that ZMP is always in the middle of moving leg, and during the walking pattern is quickly changed from one leg to the other one. Therefore, for the CoG trajectory, we can write:

$$c_x(t) = B \sum_{k=1}^{\infty} [1 - \cos \; h\omega_n(t - kT)] u(t - kT), \tag{23}$$

$$c_y(t) = A[1 - \cos \; h\omega_n(t)] + 2A \sum_{k=1}^{\infty} (-1)^k [1 - \cos \; h\omega_n(t - kT)] u(t - kT). \tag{24}$$

where $T$—step period, $A$—distance between legs and $B$—step distance in x-axis

$$S\omega_n^2 = \frac{g}{c_z}. \tag{25}$$

Equations are very unstable and sensitive for $\omega_n$ changing. The solution consists of a Fourier transformation used to describe CoG in $x$-axis and $y$-axis.

$$c_x^{ref}(t) = \frac{B}{T_0}\left(t - \frac{T_0}{2}\right) + \sum_{n=1}^{\infty} \frac{BT_0^2 \omega_n^2 (1 + \cos n\pi)}{n\pi(T_0^2 \omega_n^2 + n^2 \pi^2)} \sin \frac{n\pi t}{T_0}, \tag{26}$$

$$c_y^{ref}(t) = \sum_{n=1}^{\infty} \frac{2AT_0^2\omega_n^2(1-\cos n\pi)}{n\pi(T_0^2\omega_n^2 + n^2\pi^2)} \sin \frac{n\pi t}{T_0}. \tag{27}$$

For the calculation, we can describe:

$$x_a(t) = \begin{cases} kD_s, t = kT_c \\ kD_s + l_{an}\sin q_b + l_{af}(1-\cos q_b), t = kT_c + T_d \\ kD_s + L_{ao}, t = kT_c + T_m \\ (k+2)D_s - l_{an}\sin q_f - l_{ab}\left(1-\cos q_f\right), t = (k+1)T_c \\ (k+2)D_s, t = (k+1)T_c + T_d \end{cases}, \tag{28}$$

$$z_a(t) = \begin{cases} h_{gs}(k) + l_{an}, t = kT_c \\ h_{gs}(k) + l_{af}\sin q_b + l_{an}\cos q_b, t = kT_c + T_d \\ H_{ao}, t = kT_c + T_m \\ h_{ge}(k) + l_{ab}\sin q_f - l_{an}\cos q_f, t = (k+1)T_c \\ h_{ge}(k) + l_{an}, t = (k+1)T_c + T_d \end{cases} \tag{29}$$

where $T_c$—period for one step, $T_d$—interval at both legs statuc, $L_{ao}$ and $H_{ao}$—position of the highest ankle point on foot, $T_m$—time to reach the highest position, $D_s$—length of one step, $l_{an}$—height of foot, $l_{ab}$—length of ankle joint to the heel, $l_{af}$—length from ankle joint to the peak, $q_{gs}(k)$ and $q_{ge}(k)$—angles to the ground and $h_{gs}(k)$ together with $h_{ge}(k)$—height of foot from the ground. The first and second derivation, $x_a(t)$ and $z_a(t)$, could be continuous during the whole trajectories due to obtaining of the main trajectory. We can reach it with the help of interpolation [20].

### 3.4. Walking Pattern Simulation

In this section, we address the fundamental issues of feasibility and stability of the proposed walking model, where we verify the behavior of the model at different parameters of step length $D_s$ and the period for one step $T_c$. We performed the simulation in the CAE program Creo, with the help of the Mechanism module. The performed dynamic simulation was performed at a frequency of 100 Hz. The drives were simulated with a torque coupling, taking into account the gear ratio of the gearbox influencing its dynamic behavior.

The individual parts of the robot are made of aluminium alloy EN AW 6060, whose modulus of elasticity was specified in the Creo system. The idea of the experiment was to forward walk while changing the desired walking speed. Figure 4 shows a CAD model of the humanoid robot used for the simulation. The robot commanded a sagittal reference velocity $v_x$ 0.1 ms$^{-1}$, which is suddenly increased to 0.3 ms$^{-1}$ at the time of simulation 9 s.

During the simulation, the CoG projection data in the horizontal plane were recorded. The trajectories obtained by the simulation together with the required CoG rate are shown in Figure 5. As expected, a higher commanded speed is realized, and is achieved by increasing both the step length and the walking frequency. These parameters were generated according to Formulas (28) and (29).

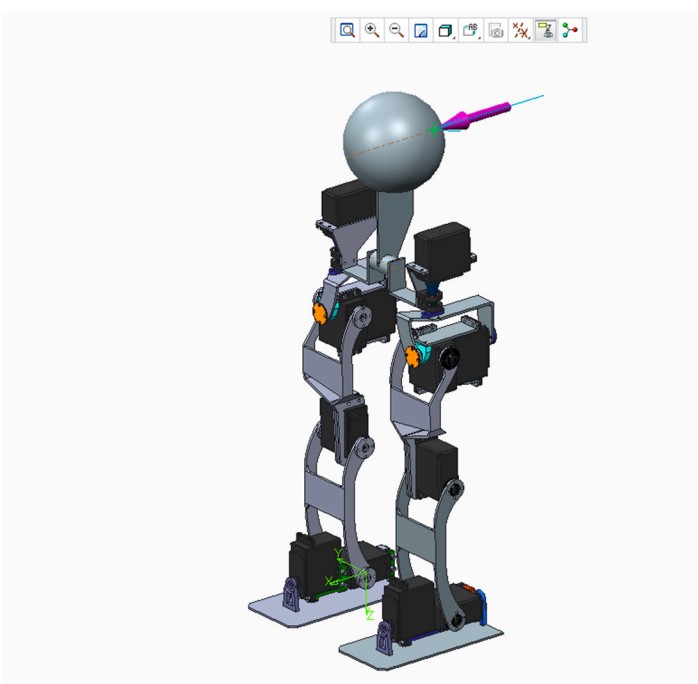

**Figure 4.** Virtual simulation of CAD model.

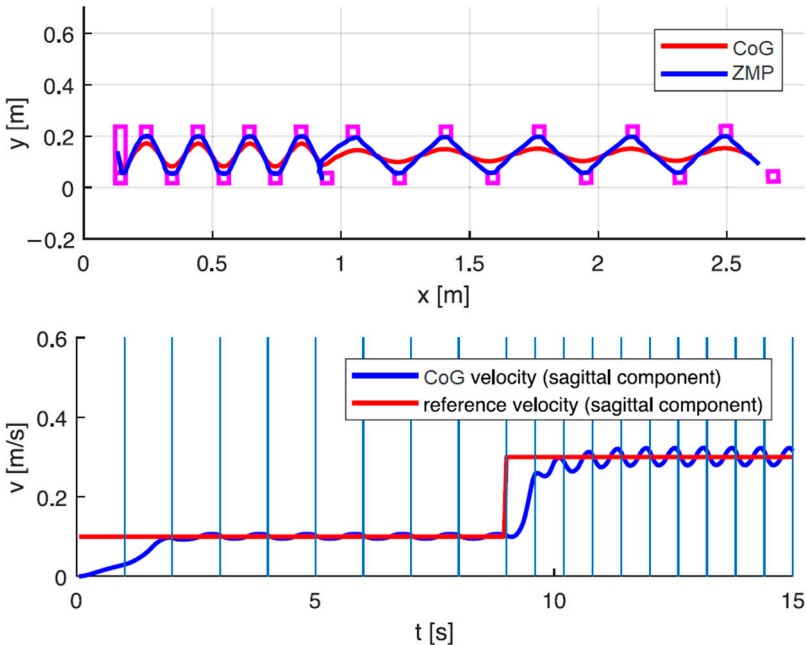

**Figure 5.** CoG motion trajectory—simulation.

## 4. Results

A walking humanoid robot was used for this experiment; see Figure 3. As we said, the foot trajectory was generated by the generating of the walking pattern using the inverted pendulum method and the foot support point. Inverse kinematics of the foot were used to calculate the angles of rotation of individual joints of the foot. The overall strategy of the walking control is described in Figure 6. The walking experiment was performed on a floor surface, which is not perfectly flat. An eight-step walk test was performed to illustrate the function. The torso positions and tilt angles of the robot were measured during the walking cycle. A position sensor was used to determine the tilt of the torso.

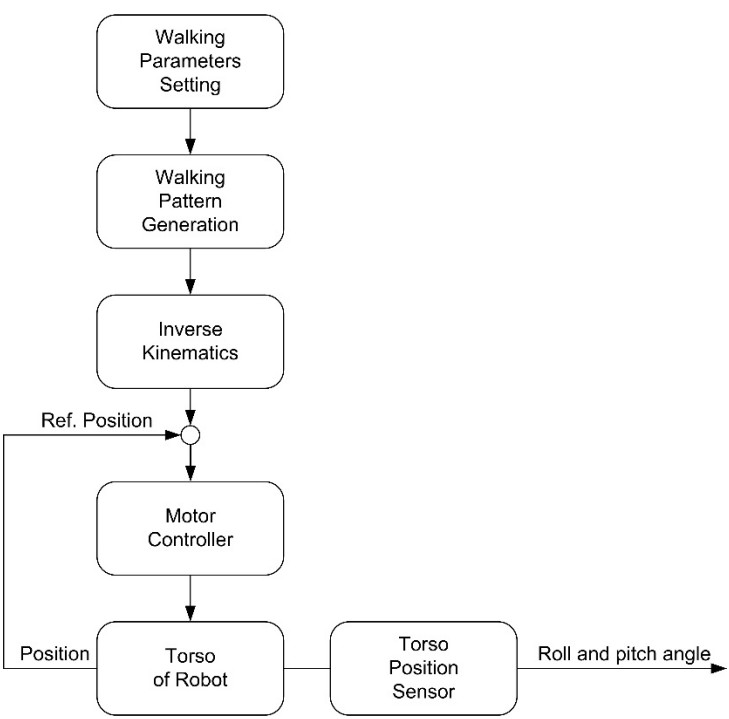

**Figure 6.** Walking strategy control.

A measured change of torso angles without any feedback, except PID engine control, is provided in Figure 7.

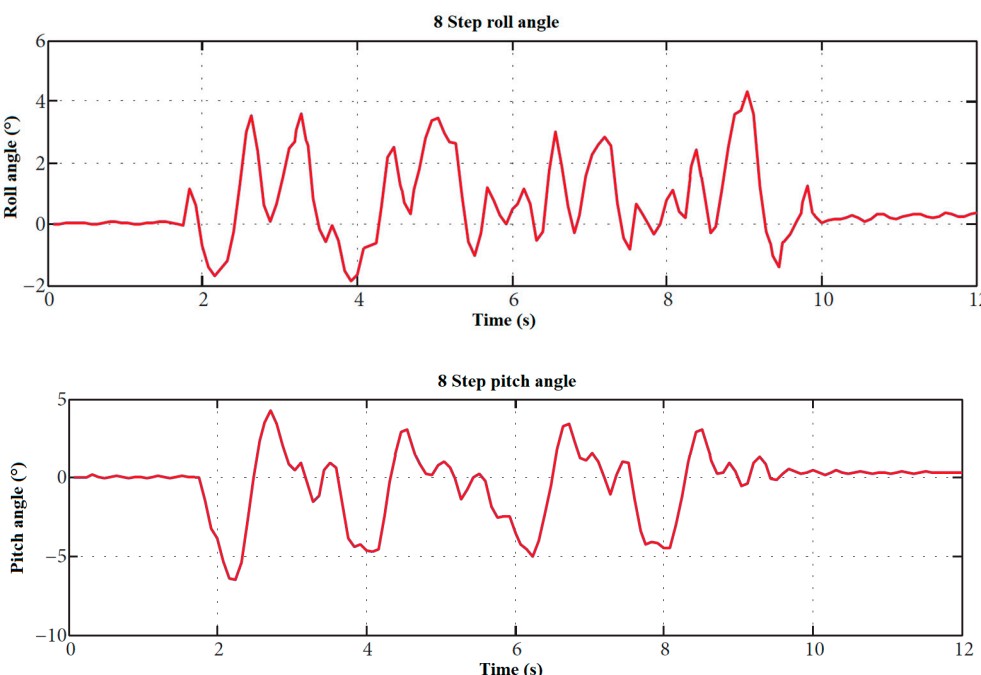

**Figure 7.** Changing of torso angles during experiment.

The aim of these experiments was to get acquainted with the basic walking algorithm and to let a two-legged robot go through several tricks. Without modified virtual restrictions, the robot could take several steps. Several experiments with walking were also performed outdoors, where the robot walked on gentle slopes and variations of the

unevenness of the ground. The robot was only able to take a few steps. The robot was able to stabilize the position only with relatively small inequalities. To use this method, it is necessary to apply additional sensors to the feet, on the basis of which it will change dynamically. This can be achieved by modifying the virtual time of the reference ZMP. Among other parameters, the walking speed was reduced. Based on the results of the performed experiments, it can be stated that the chosen method of gait generation allows for performance of complex movement maneuvers.

## 5. Discussion

These experiments confirm that the chosen concept can be largely established in activities that require a high degree of mobility in different application environments. The potential for the use of this method is there; however, our goal is to develop a robot for a robot football competition, where the robots move on a horizontal field. In football, however, other unforeseen circumstances arise, e.g., ball impact or contact with an opponent. Overall, this method is effective and robust to overcome such obstacles, which was our goal.

Current research approaches show that implementation of humanoid robots, whose kinematic chain consists of 12 DOF, is the most common solution in the field of robotics [21]. Despite several projects of bipedal projects, that have been created for purposes such as human assistance, surgery, fun and special army, there is still potential in the area of games, specifically robot soccer (robotic football). In this field, most of the bipedal robotic constructions are developed as a robotic kit with direct parts and dimensions. The basic requirement in this competition is a proportionally oriented biped robotic construction with kinematics that simulates a biological example—the human body. Our aim was focused on our own solution with specific requirements for walking cycles, stability and control [22].

Therefore, the construction of a humanoid robot is designed to be as close as possible to the biological pattern of object being analysed—the human body. Thus, the ratio between the femur and leg bone is 1:1.023, as for an average human. In our case, this length ratio is 1:1 and 110 mm in length, corresponding to a reduction of approximately 1:4 compared to the pattern. Another functional parameter is mutual spacing of right and left spherical hip joints, which is measured at the frontal plane and is 115 mm at the given kinematics. In proportion to the biological pattern, this dimension is reduced to 1:2. The foot is designed as a solid metal plate with dimensions of 120 mm × 64 mm. The average foot length in an adult is approximately 26.7 cm, which is a reduction of 1:2.225 compared to the human pattern. This was mainly done to achieve the greatest possible foot support (necessary for stable walking). The distance of the ankle joint from ground measured in a vertical plane is about 89 mm for an average adult human, or in our case 24 mm, which corresponds to a ratio of about 1:3.7.

As we can see from Figure 2, the humanoid robot consists of 12° DOF, solved as a simplified biological example for the lower limbs. Based on available knowledge from stability control methods, management strategies and human walking, we found that applying a dynamic filter and low-level joint control method would be the most appropriate management method. The essence of this motion control and together with tracking of required motion trajectories at individual joints of humanoid robot lay in formula determination from the generated rotational angle tables at individual joints, during the one-step cycle [21].

Using the ZMP criterion on a biped can achieve a dynamically stable gait. This is the reason why this criterion is widely applied on biped walking. A simulation study using the dynamical model of the robot achieved a dynamically stable step, as demonstrated.

The ZMP criterion can also be used if the robot has to perform other tasks but walking. Hence, the ZMP is a very practical criterion, though it has its limitations. In order to be stable with this criterion, the ZMP has to remain within the support polygon for all time instances. Running and jumping are never stable according to this criterion due to the fact that there is no continuous support polygon. Another limitation of walking dynamically

stable according to the ZMP criterion is that the achievable gait does not really resembles that of a real human gait.

Human-like walking is not stable according to the ZMP criterion. Nevertheless, humans do not fall while walking. If the ZMP of a human walk should be monitored, it would certainly leave the support polygon.

All of the above can be realized with the ZMP criterion. Hence, in expectation of better stability criteria or other ways to prevent the biped from falling, the ZMP is a very practical criterion to use in biped locomotion and in the posture control of bipedal robots.

The robot designed by us is designed primarily for movement on a flat surface, where the game is robotic football. This reduces the demands on gait stability. While writing the article, the robot was tested for walking rather than running. In further research, we want to continue testing the robot during a game, but in the event of a collision and loss of stability, the robot will need to be re-positioned. Therefore, the occasional loss of stability is not limiting for our robot design.

## 6. Conclusions

The most common kinematic structure with 12 DOF was chosen for the design of the robot's locomotor equipment. Therefore, after a detailed examination of movement possibilities, we conclude that this solution and proposed movement requirements will be better implemented with such a number of DOF. During the design of the humanoid robot, such considerations have been taken into account as the order of kinematic pair arrangement, which has a significant effect on system functionality as a whole. It is mainly composed of multi-axis joints, i.e., hip and ankle joints. We have achieved a more accurate imitation of human foot kinematics and simplified calculations for positions at individual joints during the creation of a mathematical model for walking cycles by using inverse kinematics. The only significant difference between the human footprint from the human foot and the proposed humanoid robot construction is the method for addressing the hip joint, which does not behave like a spherical joint, since the axis of rotation for the topmost kinematic pair is offset from the two bottoms. In the future, it is possible to design and then apply different ways of walking for this solution, which will be closer to the actual movements of human biological patterns. For this reason, the method of generating ZMP gait was chosen, as it is most similar to human gait. The data were used to build a low-cost and durable robot designed for robotic football. Our goal is to build a team of robotic humanoid football players and take part in a robotic competition organized by FIRA.

**Author Contributions:** Conceptualization, R.J., M.S. and J.S.; Data curation, Ľ.M.; Funding acquisition, R.J.; Methodology, T.K. and M.V.; Project administration, R.J.; Software, P.T. and P.M.; Writing—review & editing, M.K. and M.G. All authors have read and agreed to the published version of the manuscript.

**Funding:** This research was funded by Slovak Grant Agency KEGA: 004TUKE-4/2021 Development of innovative teaching materials for learning multi-agent robotics.

**Institutional Review Board Statement:** Not applicable.

**Informed Consent Statement:** Not applicable.

**Data Availability Statement:** Not applicable.

**Acknowledgments:** The authors would like to thank the Slovak Grant Agency 004TUKE-4/2021, the Development of innovative teaching materials for learning multi-agent robotics.

**Conflicts of Interest:** The authors declare no conflict of interest. The funders had no role in the design of the study; in the collection, analyses, or interpretation of data; in the writing of the manuscript; or in the decision to publish the results.

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
