# Peer review of "Stability and Dynamic Walk Control of Humanoid Robot for Robot Soccer Player"

_machines, doi:10.3390/machines10060463_

Round 1
Reviewer 1 Report
In general the language used need to be more precise. Authors are requested to reread and develop the text further.
There are some sentences which need to be elaborated or rewritten to make the statement more clear. One of them is "It is equipped with simple sensor structures with relatively low complexity, which is necessarily due to the resistance of the football player in the game". There are following statements which need to be explained further. for this example: resistance and low complexity seem do not have a direct relation and need clarification.
It is needed to establish a link with the statement "The deficiency of quality feedback in current robotic systems is a significant handicap in performing the technically more intricate and delicate surgical tasks inherent in specializations such as humanoid robot." with the topic of paper, i.e. stability issues.
Literature review is needed to be extended in the research field. Some of the statements have to be referenced in the first section. Even if the statement comes from the authors experience, it still need to be cited, or if the experience is from the present work, the statement shall refer to related part.
Why it is necessary to use both CoM and G together, which later become CoG? Also for equation 1, before and after same definition are repeated.
Although it is mentioned twice in "Dump elements you can see on Figure 2. You can see dump elements in Figure 2." There is no specific display of the dumping elements in Figure 2.
Table 1 needs further description in the text.
Conclusion must be improved. There are references in the conclusion. If required, they can be moved to a discussion section and conclusion must be constructed only on the findings of the research.
Author Response
Response to the comments about the paper machines-1721980
"Stability and dynamic walk control of humanoid robot for robot soccer player "
We would like to thank the Editors and the Reviewers for the evaluation of our paper, which have been very useful to improve it. We have introduced major changes to several sections. We believe that this version of the paper is much improved and we expect to have covered all mentioned problems.
Below, you will find detailed answers to reviewer's comment.
Answers to Reviewer 1
Comment R1.1
There are some sentences which need to be elaborated or rewritten to make the statement more clear. One of them is "It is equipped with simple sensor structures with relatively low complexity, which is necessarily due to the resistance of the football player in the game".
Answer to R1.1
Thanks for the comment we accepted and edited the text to make it cleaner.
Comment R1.2
It is needed to establish a link with the statement "The deficiency of quality feedback in current robotic systems is a significant handicap in performing the technically more intricate and delicate surgical tasks inherent in specializations such as humanoid robot." with the topic of paper, i.e. stability issues.
Answer to R1.2
Thank you for your feedback on maintaining stability. In the text, we have reformulated the wording of the sentence so that it is more understandable for readers.
Comment R1.3
Literature review is needed to be extended in the research field. Some of the statements have to be referenced in the first section. Even if the statement comes from the authors experience, it still need to be cited, or if the experience is from the present work, the statement shall refer to related part.
Answer to R1.3
Thank you very much for your comments. We have added a paragraph to the article on the authors' work, which dealt with a similar issue.
Comment R1.4
Why it is necessary to use both CoM and G together, which later become CoG? Also for equation 1, before and after same definition are repeated.
Answer to R1.4
We have modified the article and only use CoG, which has made the article easier to read. Thanks for the comment.
Comment R1.5
Although it is mentioned twice in "Dump elements you can see on Figure 2. You can see dump elements in Figure 2." There is no specific display of the dumping elements in Figure 2.
Answer to R1.5
Thanks for the comment, the error in the article has been corrected and the wording of the text has been modified for better readability.
Comment R1.6
Table 1 needs further description in the text.
Answer to R1.6
A more detailed description of the table has been added to the text.
Comment R1.7
Conclusion must be improved. There are references in the conclusion. If required, they can be moved to a discussion section and conclusion must be constructed only on the findings of the research.
Answer to R1.7
We have accepted your comment, we have moved part of the text to the discussion. In the end, only the facts are gained through our research.

Reviewer 2 Report
This paper is aimed to determine a walking pattern for a humanoid robot with an impact on its dynamic stability and behavior. The topic is interesting and the paper is well structured but has several issues listed below:
[1] The general impression of the manuscript is positive. The strong point of the manuscript is the importance and relevance of the topic. Negative features are the lack of a description of the relevant theoretical assumptions (some preliminary findings are listed in the introductory part, but they are not sufficient to justify the deeper theoretical context of the research) and ambiguity regarding the research sample. In general, the manuscript presents an intimate discussion of the development of a new or specific approach to assessing the readiness of the sector in question in humanoid robotics
[2] Paragraphs in Sections I, III, and others are longer. Generally, a paragraph should be at least four sentences. The basic rule for determining paragraph length is to keep each paragraph to only one main idea. If a paragraph contains multiple ideas, it is likely that the ideas aren’t fully explained or supported (in other words, the paragraph isn’t fully developed). It is hard to read the sections of the paper.
[3] The authors must add a new section called Background Literature which establishes the context of the research. This section explains why this particular research topic is important and essential to understanding the main aspects of the study. Usually, the background forms the first section of a research article/thesis and justifies the need for conducting the study and summarizes what the study aims to achieve.
[4] Overall, I have the impression that the presentation of the results is quite redundant and confusing; generally, a more concise and clear presentation should be pursued. Also, the presentation of results and interpretation of these results should not be intertwined; the latter should be presented in a discussion section.
[5] I would like to see a well-developed discussion (minimum of two pages) comparing and contrasting solutions/results presented in the work with existing work and then a subsection of it presenting contributions to theory/knowledge/literature (at least one to two paragraphs) and followed by a subsection on Implications for practice (at least one page). In these paragraphs authors should compare their research approach with previous research, citing references to others' research.
[6] The overall document should be checked for grammar, syntax, and typos errors. Based on the above comments, I strongly believe that the authors will improve the quality of their manuscript given that they will make a detailed revision of the manuscript based on the provided comments.
[7] In summary, the authors must develop sections like discussion, contribution, implication, and limitations.
Author Response
Response to the comments about the paper machines-1721980
"Stability and dynamic walk control of humanoid robot for robot soccer player "
We would like to thank the Editors and the Reviewers for the evaluation of our paper, which have been very useful to improve it. We have introduced major changes to several sections. We believe that this version of the paper is much improved and we expect to have covered all mentioned problems.
Below, you will find detailed answers to reviewer's comment.
Answers to Reviewer 2
Comment R2.1
The general impression of the manuscript is positive. The strong point of the manuscript is the importance and relevance of the topic. Negative features are the lack of a description of the relevant theoretical assumptions (some preliminary findings are listed in the introductory part, but they are not sufficient to justify the deeper theoretical context of the research) and ambiguity regarding the research sample. In general, the manuscript presents an intimate discussion of the development of a new or specific approach to assessing the readiness of the sector in question in humanoid robotics.
Answer to R2.1
Thanks a lot for the comment. We have added a paragraph to the article on the authors' work, which dealt with a similar issue.
Comment R2.2
Paragraphs in Sections I, III, and others are longer. Generally, a paragraph should be at least four sentences. The basic rule for determining paragraph length is to keep each paragraph to only one main idea. If a paragraph contains multiple ideas, it is likely that the ideas aren’t fully explained or supported (in other words, the paragraph isn’t fully developed). It is hard to read the sections of the paper.
Answer to R2.2
Thanks for the comment. We've adjusted the paragraphs to improve the readability of the article.
Comment R2.3
The authors must add a new section called Background Literature which establishes the context of the research. This section explains why this particular research topic is important and essential to understanding the main aspects of the study. Usually, the background forms the first section of a research article/thesis and justifies the need for conducting the study and summarizes what the study aims to achieve.
Answer to R2.3
Thanks for the comment. We accepted it and added an overview of works that dealt with similar issues. At the same time, we have modified the part where the novelty and contribution of the article is stated.
Comment R2.4
Overall, I have the impression that the presentation of the results is quite redundant and confusing; generally, a more concise and clear presentation should be pursued. Also, the presentation of results and interpretation of these results should not be intertwined; the latter should be presented in a discussion section.
Answer to R2.4
Thanks for the comment. Some parts of the results have been moved to the discussion.
Comment R2.5
I would like to see a well-developed discussion (minimum of two pages) comparing and contrasting solutions/results presented in the work with existing work and then a subsection of it presenting contributions to theory/knowledge/literature (at least one to two paragraphs) and followed by a subsection on Implications for practice (at least one page). In these paragraphs authors should compare their research approach with previous research, citing references to others' research.
Answer to R2.5
Part of the discussion has been extended.
Comment R2.6
The overall document should be checked for grammar, syntax, and typos errors. Based on the above comments, I strongly believe that the authors will improve the quality of their manuscript given that they will make a detailed revision of the manuscript based on the provided comments.
Answer to R2.6
The article was checked by a native speaker, and the most typos were removed.
Comment R2.7
In summary, the authors must develop sections like discussion, contribution, implication, and limitations.
Answer to R2.7
We accept the comment, thank you. We have modified the article according to your comments, which has definitely increased the informative value of the article.

Round 2
Reviewer 1 Report
the manuscript can be accepted as in the current form.
Reviewer 2 Report
The authors addressed all my concerns in this revised version. I appreciate them a lot for their effort.